# Population structure of three New Zealand crested penguins identifies current conservation challenges for the Fiordland penguin/tawaki, erect-crested penguin, and eastern rockhopper penguin

Jeff White[1,2¤]*, Philip Lavretsky[3], Pablo Garcia Borboroglu[4,5,6], Alexis Díaz[1,7], Ursula Ellenberg[4,6,8], David Houston[6,9], Robin Long[10], Thomas Mattern[4,6,11], Herman L. Mays[2], Klemens Pütz[12], Philip J. Seddon[11], Kevin G. McCracken[1,13,14]

1 Department of Biology, University of Miami, Coral Gables, Florida, United States of America,
2 Department of Biology, Marshall University, Huntington, West Virginia, United States of America,
3 Department of Biological Science, University of Texas El Paso, El Paso, Texas, United States of America, 4 Global Penguin Society, Puerto Madryn, Chubut, Argentina, 5 Centro Nacional Patagónico-(CONICET), Puerto Madryn, Chubut, Argentina, 6 The Tawaki Trust, Dunedin, New Zealand, 7 División de Ornitología, Centro de Ornitología y Biodiversidad (CORBIDI), Lima, Perú, 8 Department of Marine Sciences, University of Otago, Dunedin, New Zealand, 9 Department of Conservation, Auckland, New Zealand, 10 West Coast Penguin Trust, Hokitika, New Zealand, 11 Department of Zoology, University of Otago, Dunedin, New Zealand, 12 Antarctic Research Trust, Bremervörde, Germany, 13 Department of Marine Biology and Ecology, Rosenstiel School of Marine, Atmospheric, and Earth Science, University of Miami, Miami, Florida, United States of America, 14 Human Genetic and Genomics, University of Miami Miller School of Medicine, Miami, Florida, United States of America

¤ Current affiliation: Department of Public and Ecosystem Health, Cornell University College of Veterinary Medicine, Ithaca, New York, United States of America
* jeffwwhite90@gmail.com

## Abstract

Identifying contemporary population structure and genetic connectivity among sea-bird populations is essential for developing conservation plans for threatened species, especially as factors like philopatry, non-breeding behavior, and oceanographic features might limit gene flow between isolated populations and influence changes in genetic diversity over time. Here, we characterize the population structure of three closely related crested penguin species in New Zealand: Tawaki (*Eudyptes pachyrhynchus*; Fiordland penguins), erect-crested penguins/tawaki nana hī (*Eudyptes sclateri*), and eastern rockhopper penguins/tawaki piki toka (*Eudyptes filholi*). Whereas tawaki populations appear to be stable, the erect-crested and eastern rockhopper penguin populations have seen dramatic declines in the recent historical record. To understand the genetic implications of these differences in population trajectories, we assessed genetic connectivity among multiple colonies using thousands of nuclear autosomal loci. Our results indicate that tawaki are a single, genetically diverse population without colony-based structure, which is consistent with the currently observed stable or increasing population of tawaki. However, conservation

**Data availability statement:** All sequence data has been uploaded to the NCBI SRA repository and are available under BioProject ID PRJNA1277195.

**Funding:** 1.Vontobel Foundation (Award Von_1-2022_ART) – KP, TM https://www.vontobel.com/. 2.James A. Kushlan Endowment for Waterbird Biology and Conservation at the University of Miami – KGM https://news.miami.edu/as/stories/2013/09/kevin-g-mccracken-named-inaugural-kushlan-chair-in-waterbird-biology-and-conser.html. 3.National Science Foundation (Award #2322123) – HLM https://new.nsf.gov/funding. 4.Antarctic Research Trust – KP https://antarctic-research.de/wordpress2014/?page_id=1900&lang=en. 5.Global Penguin Society – PGB https://www.globalpenguinsociety.org/. 6.Royal Naval Birdwatching Society – JW https://www.rnbws.org.uk/. 7.Birds New Zealand Research Fund (BNZRF) – JW https://www.birdsnz.org.nz/research/. 8.Shearwater Foundation – JW https://shearwaterfoundation.org/. 9.Patreon Supporters of the Tawaki Project – TM, UE https://www.patreon.com/TawakiProject. The funders had no role in study design, data collection and analysis, decision to publish, or preparation of the manuscript.

**Competing interests:** The authors have declared that no competing interests exist.

efforts should continue to prioritize protecting marine habitats to safeguard this species. In contrast, we identified two genetically distinct populations of erect-crested penguins corresponding to the Antipodes Islands and the Bounty Islands groups. The Antipodes Islands eastern rockhopper population exhibited high levels of coancestry and low genetic diversity, consistent with population decline and limited immigration. The lack of gene flow and genetic diversity in both erect-crested and eastern rockhopper penguins on the Antipodes Islands raises concerns and highlights the need for continued research to identify the causes of declines to inform conservation efforts of these penguins.

## Introduction

Wildlife conservation management requires an understanding of demographic histories and population interconnectedness (i.e., gene flow), as both are essential for addressing genetic diversity declines in the face of ongoing population size changes in increasingly fragmented and altered ecosystems [1–3]. Preserving connectivity between populations of wide-ranging species, such as seabirds, is therefore crucial for effective management and conservation efforts. Seabirds are among the most threatened groups of vertebrates worldwide, with up to 70% of seabird species currently declining [4,5]. This is thought to be primarily due to the negative impacts of invasive species, fisheries bycatch, and climate change [6]. In particular, penguins around the world are experiencing dramatic declines due primarily to climate-driven environmental changes at sea, industrial fisheries, and human impacts on land [6–9]. Therefore, understanding the ecology and evolutionary history of penguins through studies of population genetic structure is required, as these types of studies reveal past demographic histories and may identify populations of conservation concern [10].

Although seabirds are generally highly mobile and capable of covering large distances at sea, most seabird species exhibit strong philopatry to breeding sites [11]. This can be an isolating mechanism that creates barriers to gene flow between populations [12] and potentially reduces a population's ability to recover from catastrophic events [13–15]. The dispersal of seabirds is also limited by hydrography [16]. In particular, frontal zones separating water masses have been identified as significant barriers to gene flow in some penguin species [17–19]. Such oceanographic barriers can reduce overlaps in foraging areas among populations during both breeding and non-breeding periods. Over time, this may result in allochrony within the annual cycle as populations shift the timing of events (i.e., breeding, non-breeding dispersal) to coincide with optimal foraging conditions within their preferred foraging area [20–23]. In some cases, allochrony and marine habitat segregation can eventually lead to speciation [24]. However, high dispersal capacity can decrease genetic structure across populations of wide-ranging seabirds [25].

### Study system

Our study focuses on understanding the genetic population structure and connectivity of three crested penguin species from New Zealand. The crested penguins of New

Zealand present an interesting opportunity to compare the population structure of closely related species with very different dispersal patterns and population trajectories. Specifically, the Fiordland penguin (*Eudyptes pachyrhynchus*; hereafter referred to by its te reo Māori name 'tawaki') was once thought to be in decline [26], but recent surveys have indicated that populations of this species may be stable or even increasing [27–29] and expanding in range [30]. Tracking studies have shown that during the non-breeding season, tawaki are highly mobile and cover vast distances between the Subtropical Front (STF) and the Subantarctic Front (SAF) to forage in subantarctic waters [31]. Previous studies using single-locus mitochondrial DNA indicated that the tawaki population has remained stable with continued gene flow over the last 1,151 years [32]. However, finer scale gene flow among individual colonies has not been assessed using modern DNA sequencing methods that incorporate many loci. Reduced-representation sequencing methods, such as RADseq, are well suited to evaluating broad-scale patterns of population structure, particularly in species with shallow divergence [33].

In contrast, the crested penguin species inhabiting New Zealand's Subantarctic islands have been in decline for decades. For example, eastern rockhopper penguins (*Eudyptes filholi*, 'tawaki piki toka') have declined by 94% on Campbell Island since the 1940s [34]. On the Antipodes Islands, the eastern rockhopper penguin population dropped by 92% over a 22-year period [35,36]. A recent survey of the Orde Lees colony on Antipodes Island reported a 46% decline since 2011 alone [37]. This decline in eastern rockhopper penguins in New Zealand is thought to be primarily a result of changing marine conditions leading to alterations in prey availability [35,36,38,39].

The erect-crested penguin (*Eudyptes sclateri*, 'tawaki nana hī') is currently found only on the Antipodes Islands and in the Bounty Islands group. Erect-crested penguins show strikingly different population trends between island groups. On the Antipodes Islands the breeding population of erect-crested penguins has declined by 29–42% over the last decade [36,37], yet on the Bounty Islands, erect-crested penguins have remained relatively stable over the same period [40], such that the Bounty Islands are now presumed to host most of the breeding population [37]. The two populations of erect-crested penguins exhibit slight breeding allochrony, with the Bounty Islands population starting the breeding season 2–3 weeks later than on the Antipodes Islands [41]. The populations also have differing foraging behavior during the pre-molt period. The Antipodes Islands penguins remain south of the STF during the pre-molt, while those from the Bounty Islands forage along the STF itself nearer to their breeding colonies [42]. The diverging dispersal patterns during the pre-molt period suggests that the STF influences penguin movements, thus driving the segregation between the Bounty and Antipodes Islands populations.

To understand population genetic structure of these penguins, we explored whether population structure exists among and within tawaki, eastern rockhopper, and erect-crested penguins. We utilized genome-wide, single nucleotide polymorphism (SNP) data to compare range-wide patterns of genetic diversity among and within these species to characterize population structure, including the degree of connectivity between intraspecific colonies. Among species, we were interested in comparing the different evolutionary histories of each taxon as it relates to the demography and biogeography of each. For tawaki we were interested in establishing the degree of gene flow among breeding colonies, if there is any, and determining whether there is genetic evidence that this species indeed represents a single, panmictic population using different genetic markers than previous studies. In contrast to tawaki, in which we expected to find a high degree of genetic diversity, we expected low genetic diversity among eastern rockhopper penguin populations of the Antipodes Islands in light of their alarming declines. Finally, we discuss whether the presence of the STF, and associated higher primary productivity, influences gene flow between the Antipodes Islands and Bounty Islands populations of erect-crested penguins.

## Materials and methods

### Sample collection

We collected blood samples (1 mL) from the brachial vein of adult tawaki (*n* = 55 individuals) across five colonies during the breeding seasons (September-October) of 2017, 2018, and 2022. Our sampling was divided into four regions

comprising five sites: South Westland (Jackson Head, $n = 13$) Milford Sound (Harrison Cove, $n = 15$), Doubtful Sound (East Shelter Island, $n = 13$; Seymour Island, $n = 7$), and Foveaux Strait (Whenua Hou, $n = 7$) (Fig 1; S1 Table). The sex of all tawaki was determined in the field using published morphometric characters [43]. We then collected blood samples from erect-crested and eastern rockhopper penguins in November-December 2022 [42] (S1 Table). Erect-crested penguins were sampled from the north and south coasts of Antipodes Island ($n = 28$) as well as on Proclamation Island ($n = 26$) in the Bounty Islands group (Fig 1). Eastern rockhopper penguins were sampled on the north and south sides of Antipodes Island ($n = 17$) only, as they do not occur in the Bounty Islands (Fig 1). We determined the sex of erect-crested and eastern rockhopper penguins in the field following Davis et al. [44] and Warham [45], respectively. All samples collected in 2017 and 2018 were stored in 80% ethanol, whereas those collected in 2022 were stored in a blood lysis buffer [46]. Upon returning to the lab, all samples were frozen and stored at –80°C.

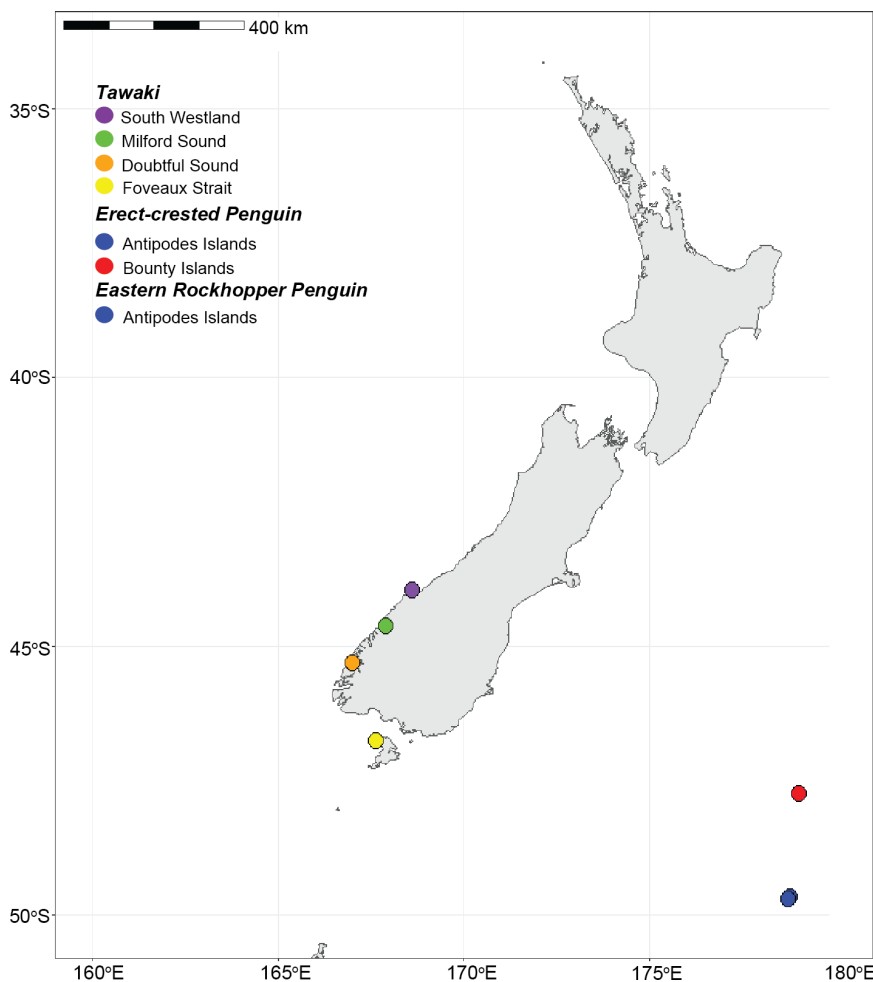

**Fig 1. Map of all sampling locations.** All sampling occurred in southern New Zealand during the breeding seasons of 2017, 2018, and 2022. Colored points represent sampling locations for tawaki (purple, green, orange, yellow; $n = 55$), erect-crested penguins (red, blue; $n = 54$), and eastern rockhopper penguins (blue; $n = 17$).

                                                                          

## DNA extraction and ddRAD-Seq library preparation

We extracted DNA from each blood sample either using a DNeasy Blood and Tissue kit (2022 samples) following the standard manufacturer protocols (Qiagen, Valencia, CA, USA) or with phenol-chloroform-isoamyl alcohol extractions (2017 and 2018 samples). We then performed partial genome sequencing via double-digest restriction-associated DNA (ddRAD) sequencing methods following protocols of DaCosta and Sorenson [47], but with size-selection following Hernández et al. [48]. In short, genomic DNA was first enzymatically digested using SbfI and EcoRI restriction enzymes followed by adapter ligation with Illumina TrueSeq compatible reagents and barcodes for demultiplexing. Next, we conducted double-sided, bead-based size selection of adapter-ligated DNA fragments [49]. Finally, clean libraries were multiplexed in equimolar amounts and sent out for sequencing on an Illumina HiSeq X using single-end 150 bp chemistry at Novogene Ltd. (Sacramento, CA, USA).

## Bioinformatics of ddRad-Seq data

All raw Illumina reads were first demultiplexed using the ddRADparser.py python script of the DaCosta and Sorenson (http://github.com/BURAD-seq/ddRad-seq-Pipeline) pipeline [47]. Next, all reads were cleaned using Trimmomatic v0.36v [49] before being aligned to the macaroni penguin (*Eudyptes chrysolophus)* reference genome (https://www.ncbi.nlm.nih.gov/datasets/genome/GCA_010084205.1/) using Burrows Wheeler Aligner v.0.7.1.7 [50]. Samples were then sorted and indexed in Samtools v.1.6 [51] and combined and genotyped using bcftools v.1.6 (as part of the SAMtools package) "mpileup" and "call" functions with the following parameters "-c –A -Q 30 -q 30," which set a base pair and overall sequence PHRED score of ≥30 to ensure that only high-quality sequences are retained. Note all steps through genotyping were automated using in-house Python scripts (https://github.com/jonmohl.PopGen; [52]). Next, we used VCFtools v. 0.115 [53] to further filter VCF files for any base-pair missing >20% of samples, which also required a minimum base-pair sequencing depth coverage of 5X (i.e., 10X per genotype) and quality per base PHRED scores of ≥30 to be retained.

## Population structure

Population structure analyses were based on autosomal ddRAD-seq biallelic single nucleotide polymorphisms (SNPs) only. We used PLINK v.1.90 [54] to ensure that singletons and any SNP missing >20% of data across samples were excluded in each dataset. Additionally, we identified independent SNPs by conducting pairwise linkage disequilibrium (LD) tests across ddRAD-seq autosomal SNPs (--indep-pairwise 2 1 0.5) in which 1 of 2 linked SNPs are randomly excluded if we obtained an LD correlation factor ($r^2$) > 0.5. All analyses were done without *a priori* information on population or species identity. Note these steps were done for each separate analysis that included (1) all penguins, (2) tawaki only, (3) erect-crested penguins only, and (4) eastern rockhopper penguins only.

Next, we used the PCA function in PLINK to perform a principal component analysis (PCA). ADMIXTURE v.1.3 [55,56] was used to attain maximum likelihood estimates of population assignments for each individual, with datasets formatted for the ADMIXTURE analysis using PLINK v.1.90, and following steps outlined in Alexander et al. [55]. We ran each ADMIXTURE analysis with a 10-fold cross validation, incorporating a quasi-Newton algorithm to accelerate convergence [57]. Each analysis used a block relaxation algorithm for point estimation and terminated once the change in the log-likelihood of the point estimations increased by <0.0001. Each analysis was run for $K$ population values with standard-errors derived from 100 bootstrap replicates per each value of $K$. We ran $K = 1–10$ for the all-species analyses and $K = 1–5$ for species-specific analyses. The optimum $K$ in each analysis was based on cross-validation errors per $K$ value; however, we examined additional values of $K$ to test for further structural resolution across analyses. Finally, we also visualized coancestry assignments using fineRADstructure v.0.3 [58,59]. This method infers a matrix of coancestry coefficients based on the distribution of identical or nearest neighbor haplotypes among samples. Coancestry at each locus is divided equally among all individuals with identical haplotypes, or in the case of a unique allele, with the nearest neighbor

haplotype [59]. Rare haplotypes, as characterized by rare SNPs, are on average of more recent origin [60] and contribute the most to the coancestry index. This then provides a measure that highlights recent coancestry. We completed a burn-in of 100,000 iterations, followed by 100,000 Markov chain Monte Carlo iterations. Finally, we constructed the phylogenetic tree using the software's default parameters.

## Summary statistics

Summary statistics including pair-wise estimates of relative genetic divergence (pairwise $\Phi_{ST}$) and average per group nucleotide diversity (π) were calculated among all three species in VCFtools. We also calculated π and Tajima's D among locations within the tawaki and erect-crested penguin samples. For the erect-crested penguins, we were interested in the connectivity of the Antipodes Islands and Bounty Islands groups and therefore calculated these statistics between the two island groups as well.

## Changes in effective population size over time

Long-term demographic changes were inferred using single-population ∂a∂i models that estimate the effective population size ($N_e$) over time, as detailed by Hernández et al. [48]. We utilized all recovered ddRAD-seq autosomal loci to create a one-dimensional site-frequency spectrum (SFS) for each population, transforming Nexus-formatted SNP datasets into population-specific SFS using custom python scripts (all developed scripts are available here: https://github.com/jibrown17/Dove_dadi.demographics). This model, designed for continuous $N_e$ estimation, implemented a stepwise time interval function and conducted 100 iterations with ∂a∂i's ''Integration.one_po'' function, effectively modeling $N_e$ transitions from ancestral levels to current $N_e$ estimates across sequential time intervals. This method incorporates a rigorous model-fitting phase, during which comparisons are made across 50 runs for each population to solidify the robustness of the $N_e$ calculations. We calibrated the final optimal parameters against empirical data using a mutation rate-derived scale factor ($\theta = 4N_{Anc} \times \mu$, where $N_{Anc}$ represents ancestral $N_e$, and μ represents the mutation rate). We assessed model accuracy based on log-likelihood comparisons with empirical data and calculated the 95% confidence intervals (CIs) using the parameter uncertainty metrics included in ∂a∂i. The $N_e$ and time parameters were converted into biologically informative values, as previously described, using generation time ($T_g$), calculated as a function of the age of sexual maturity (α = 5 years) and survival (s = 0.89), where $T_g = \alpha + (s/1 - s)$ [32]. We assumed a substitution rate $\mu = 1.91 \times 10^{-9}$, calibrated with divergence estimates obtained from prior demographic analyses [61] with the corresponding number of total base pairs sequenced and passing all threshold tests as 432,989 bp for tawaki, 375,754 bp for erect-crested penguin, and 413,822 bp for rockhopper penguin.

## Ethics statement

This project was approved by the Marshall University Office of Research Integrity's Institutional Animal Care and Use Committee (IACUC) under protocol #686, the University of Miami Institutional Animal Care and Use Committee (IACUC) under protocol #20–090, and the University of Otago's Animal Ethics Committee (AUP D69/17). All field work and permissions were granted under the Department of Conservation (DOC) permit authorization numbers-FAU 86101-FAU and 78612-FAU.

## Results

### Population structure

A total of 436,292 base-pairs (bps) met threshold criteria, with average individual sequencing depth of 71X (range = 7–113X). For the population structure analyses, we had a total of 19,974 (of 21,513 and 97% of alleles present), 8,953 (of 9,991 and 96% of alleles present), 5,018 (of 5,760 and 99% of alleles present), and 5,213 (of 7,173 and 99% of alleles present) SNPs for all-species combined, erect-crested penguin, tawaki, and eastern rockhopper analyses, respectively. Concordant with

the ADMIXTURE analysis, we found $K=3$ populations to be the optimum model corresponding to the species limits (Table 1). Furthermore, the same three genetic clusters were recovered in both the PCA and fineRADstructure coancestry analysis (Figs 2 and 3). No subpopulation structure was found among tawaki populations for which the coancestry assignments (Fig 3) and ADMIXTURE analyses found an optimum model of $K=1$ within this species (Table 1). Conversely, both the PCA (Fig 4) and ADMIXTURE assignment probabilities (Fig 5) revealed two distinct genetic clusters corresponding to erect-crested penguins from the Antipodes and Bounty Islands group.

## Summary statistics

Pairwise $\Phi_{ST}$ revealed the highest differentiation between tawaki and eastern rockhopper penguins ($\Phi_{ST}=0.28$; Table 2) and lower differentiation between erect-crested penguins and tawaki ($\Phi_{ST}=0.17$; Table 2) and eastern rockhopper penguins ($\Phi_{ST}=0.16$; Table 2), respectively. Pairwise $\Phi_{ST}$ was lowest between the two island groups of erect-crested penguins ($\Phi_{ST}=0.01$). Next, among the three penguin species, we recovered the lowest coancestry among tawaki, followed by erect-crested penguins (Fig 5). Eastern rockhopper penguins exhibited the highest coancestry, including a mated pair that are likely full siblings (Fig 5). This high degree of coancestry suggests a limited gene pool likely due to historical

**Table 1. Cross validation (CV) and cluster assignment (K).**

| Group | CV | K |
|---|---|---|
| All samples | 0.37658 | 1 |
| | 0.24999 | 2 |
| | **0.17026** | **3** |
| | 0.17818 | 4 |
| | 0.18734 | 5 |
| | 0.19054 | 6 |
| | 0.1953 | 7 |
| | 0.20426 | 8 |
| | 0.21649 | 9 |
| | 0.21921 | 10 |
| Tawaki | **0.44067** | **1** |
| | 0.47012 | 2 |
| | 0.51374 | 3 |
| | 0.54956 | 4 |
| | 0.61122 | 5 |
| Erect-crested penguin | **0.41608** | **1** |
| | 0.45379 | 2 |
| | 0.49941 | 3 |
| | 0.54918 | 4 |
| | 0.61152 | 5 |
| Eastern rockhopper penguin | **0.69060** | **1** |
| | 0.92759 | 2 |
| | 1.26146 | 3 |
| | 1.50218 | 4 |
| | 1.48068 | 5 |

Cross validation (CV) and cluster assignment (K) for ddRAD-Seq data of all samples pooled as well as each species calculated individually. The optimal K assignment is presented in bold for each group.

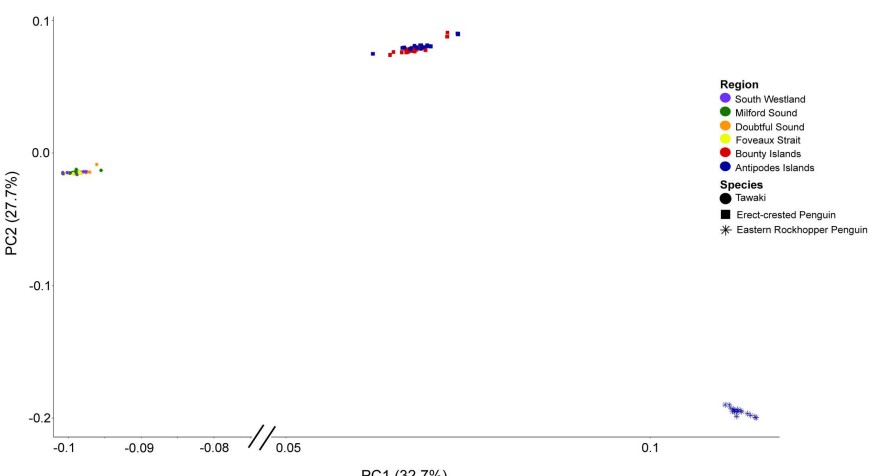

**Fig 2. Principal components analysis (PCA) of all ddRAD-seq data.** Tawaki (circle), erect-crested penguins (square), and eastern rockhopper penguins (asterisk) colors denote the region in which the samples were collected. The first two principal components PC1 and PC2 represent 60.4% of the variance and separate the samples into three distinct genetic clusters.

population declines (Table 2). Tajima's D revealed a positive and higher value in tawaki ($D = 0.09$) than in either the erect-crested penguins ($D = -0.31$) or eastern rockhopper penguins ($D = -0.44$; Table 2), which were both negative. Tajima's D was less negative in the Antipodes Islands ($D = -0.16$) versus the Bounty Islands ($D = -0.32$) populations of erect-crested penguins (Table 2).

### Changes in effective population size over time

Analyses of historical changes in effective population size ($N_e$) showed divergent trends between the temperate and sub-antarctic species. Tawaki $N_e$ has remained relatively stable over the last 200,000 years followed by a very recent increase in numbers close to the present (Fig 6a). $N_e$ estimates for eastern rockhoppers show evidence of a continuous decline over the last 100,000 years to the present (Fig 6b), as do both the Antipodes Islands (Fig 6c) and Bounty Islands (Fig 6d) populations of erected-crested penguins. Historically, $N_e$ was higher for the erect-crested penguins than for the eastern rockhopper penguins.

## Discussion

### Tawaki represent a panmictic population

Tawaki are known to be philopatric [62] and both socially and reproductively monogamous [63], which is expected to result in population structure [11,17,18]. However, our population structure analysis does not support genetic differentiation among colonies as a result of philopatry or monogamy. Instead, it confirms that tawaki represent a single panmictic population with all pairwise $\Phi_{ST}$ comparisons between tawaki colonies equal to zero. This could potentially be explained by lower philopatry than expected or as shown, recent population expansions, each of which might homogenize genetic diversity and masking underlying demographic structure [64,65]. However, Cole et al. [32] analyzed cytochrome c oxidase (COI) and the mitochondrial control region (CR) of tawaki and found no evidence of population structure alongside consistent gene flow over the last 1,151 years. Therefore, masking of population structure by the observed recent population expansions is unlikely in tawaki. Our estimates of nuclear genetic diversity support these prior findings of mitochondrial DNA diversity and, together, provide a clearer view of the interconnectedness of tawaki colonies across their range (Fig 5). Estimates of historical changes to effective population size ($N_e$) furthermore support the idea of a relatively stable

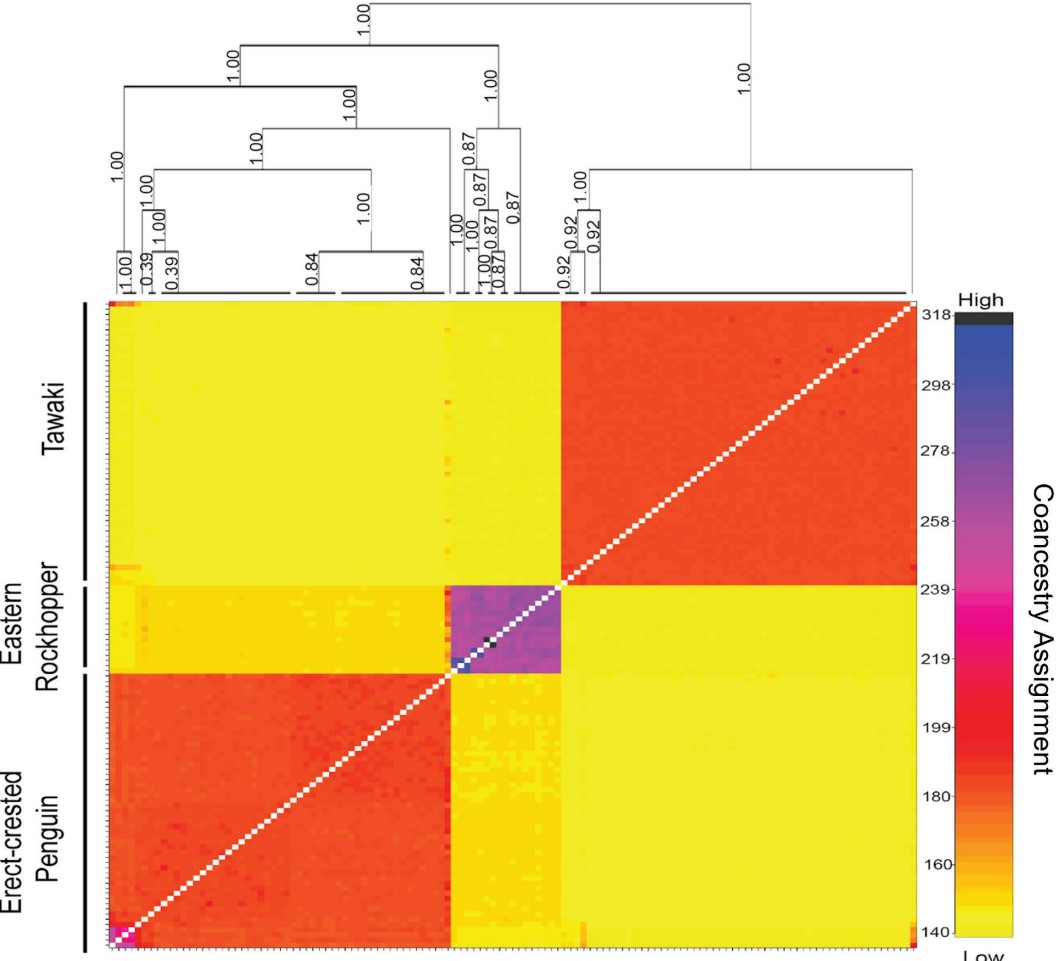

**Fig 3. fineRADstructure heat map.** Heat map resulting from the population assignment analysis performed using fineRADstructure on all ddRAD-Seq data combined. Tick marks in both horizontal and vertical axes represent individual samples. The coancestry matrix shows the pairwise genetic similarity between individuals. Inferred populations are indicated by the dendrogram on the right. Overall, tawaki exhibit the lowest degree of coancestry whereas the eastern rockhopper penguins exhibit very high coancestry.

population over time with evidence of more recent range expansions (Fig 6a). Although tawaki may have undergone range reductions since the arrival of early Polynesians, the Fiordland region in southwestern New Zealand has likely provided a crucial refugium harboring a large population with high levels of genetic diversity [32]. Therefore, continued monitoring and management of this core region of their range will be essential for the conservation of tawaki.

## Conservation concerns for the eastern rockhopper and erect-crested penguins

Overall, our data corroborate a concerning trend for the eastern rockhopper and erect-crested penguins. Both species have been experiencing significant declines on the Antipodes Islands over the last several decades, albeit at different rates [36,37]. The decline in eastern rockhopper penguins on the Antipodes Islands has implications for the persistence of the species in the region. Our results suggest that there is a very high degree of coancestry among the eastern rock-hopper penguins on Antipodes Island. In fact, one breeding pair we sampled appears to be full siblings (Fig 5). Population bottlenecks are often cited to explain high levels of coancestry within a population [66–68]. However, the reduction

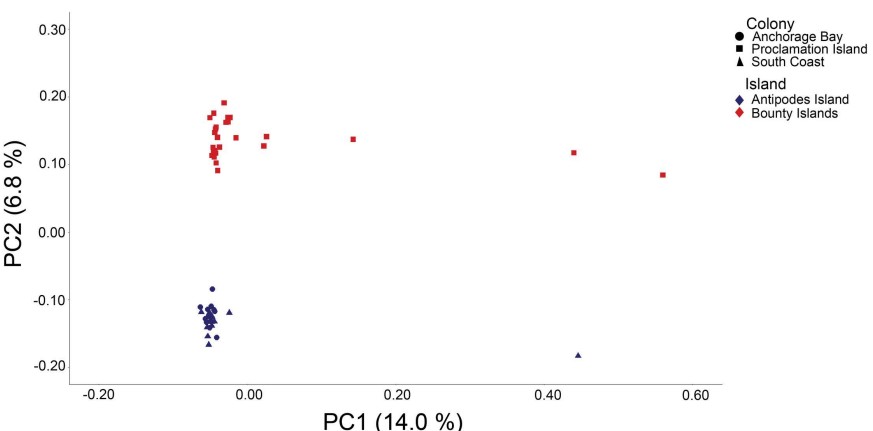

**Fig 4. Principal components analysis (PCA) for erect-crested penguins.** All ddRAD-seq data for erect-crested penguins from the Anchorage Bay (circle), South Coast (triangle), and Proclamation Island (square) colonies. The first two principal components PC1 and PC2 represent 20.8% of the variance and separate the samples into two distinct genetic clusters: Antipodes Island (blue) and Bounty Islands (red). Note six individuals that fall outside of the main clusters.

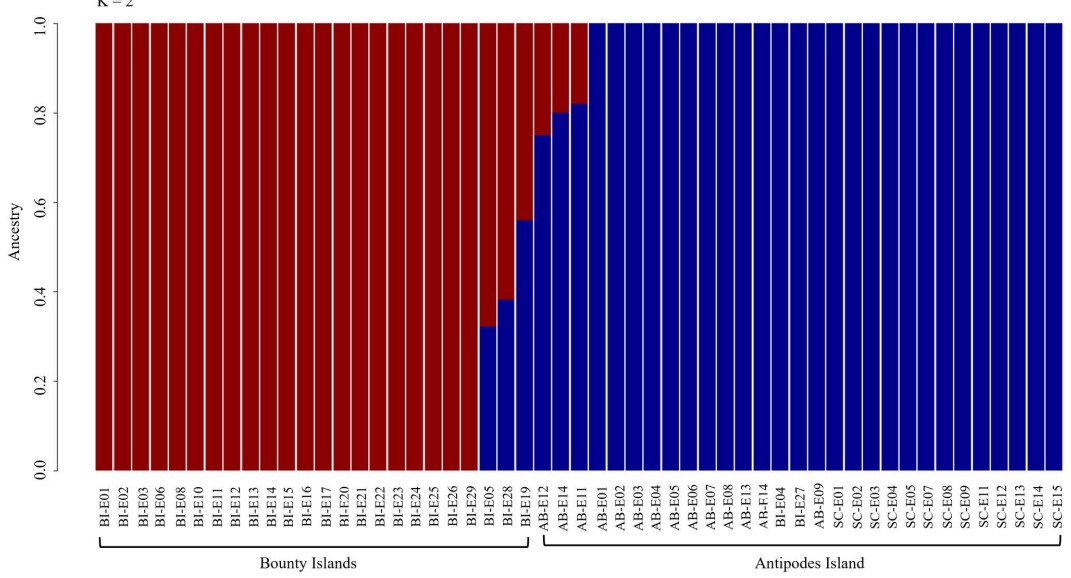

**Fig 5. ADMIXTURE plot for erect-crested penguins.** ADMIXTURE plot from ddRAD-Seq data from erect-crested penguins showing assignments to population 1 or 2 based on $K=2$. Six individuals including three from the Bounty Islands and three from the Antipodes Islands and falling outside the clusters on the PCA appear as 'hybrids' between the two groups.

in population size and genetic diversity following a bottleneck would be expected to rebound with even modest levels of immigration [69–71]. For example, king penguins (*Aptenodytes patagonicus)* on Macquarie Island recovered pre-bottleneck levels of genetic diversity within 80 years of a halt in human exploitation [72]. We suggest that the low level of genetic diversity in this subset of the eastern rockhopper population is indicative of an extended decline in the breeding population on the Antipodes Islands, which has not recovered, leaving the remaining individuals fewer options for out-crossing. This coincides with the rise of global temperatures over the 20th century as has been previously implicated in

**Table 2. Summary statistics for ddRAD-Seq data.**

| Group | π | Tajima's D | $\Phi_{ST}$ |
|---|---|---|---|
| Tawaki | 0.0021 | 0.09 | – |
| Erect-crested penguin (all) | 0.0019 | −0.31 | – |
| Erect-crested penguin (Antipodes) | 0.0022 | −0.16 | – |
| Erect-crested penguin (Bounty) | 0.0021 | −0.32 | – |
| Eastern rockhopper | 0.0023 | −0.44 | – |
| Tawaki vs. Erect-crested penguin | – | – | 0.17 |
| Tawaki vs. Erect-crested penguin (Antipodes) | – | – | 0.22 |
| Tawaki vs. Erect-crested penguin (Bounty) | – | – | 0.21 |
| Tawaki vs. Eastern rockhopper | – | – | 0.28 |
| Erect-crested penguin Antipodes vs. Bounty | – | – | 0.01 |
| Erect-crested penguin vs. Eastern rockhopper | – | – | 0.16 |
| Erect-crested penguin (Antipodes) vs. Eastern rockhopper | – | – | 0.22 |
| Erect-crested penguin (Bounty) vs. Eastern rockhopper | – | – | 0.16 |

Summary statistics for ddRAD-Seq data collected on tawaki, erect-crested penguin, and eastern rockhopper penguins including pairwise $\Phi_{ST}$, nucleotide diversity (π), and Tajima's D.

the decline of eastern rockhoppers in the region [38,39]. However, it must also be noted that our estimates of historical trends in $N_e$ suggest that eastern rockhopper populations may have been in slow decline for much longer (Fig 6b). Our analysis also suggests a lack of gene flow to the Antipodes Islands from other colonies in the region such as those on Auckland or Campbell Islands, further reducing chances of genetic recovery. Sampling of these other eastern rockhopper populations is crucial for evaluating gene flow and connectivity in this species. Overall, the eastern rockhoppers on the Antipodes Islands may be experiencing a contemporary bottleneck due to these sharp declines. While the exact causes of this decline are not fully known, changing marine conditions and shifting prey abundance and distribution are likely contributing factors [36]. Further research should be undertaken to characterize the degree of coancestry within and between the Antipodes, Campbell, and Auckland Island populations to better understand the magnitude of our findings for the species as a whole in the region.

The erect-crested penguin presents a unique case among New Zealand penguins. It is found only on the Antipodes and Bounty Island groups, yet the population trajectories are different in each location. In the mid-1990s, the Antipodes held around 66% of the world's erect-crested penguin population, whereas the most recent surveys indicate the Bounty Islands will become the new species stronghold before 2030 (Mattern et al., in prep.). Historical trends in $N_e$ furthermore suggest a general decline over the last 100,000 years (Fig 6c-d). As for the eastern rockhopper penguins, the cause of the decline of erect-crested penguins on the Antipodes Islands is thought to be due primarily to changing marine conditions and, thus, prey abundance [36]. Although only weakly differentiated in allele frequencies (Table 2), our results suggest that erect-crested penguins comprise two distinct subpopulations corresponding to the Antipodes Islands and Bounty Islands groups (Figs 4 and 5). The small proportion of the overall Bounty Islands population that had Antipodes Islands ancestry suggests that the more stable population of the Bounty Islands is not being significantly augmented by immigration from the Antipodes Islands (Figs 3 and 4; Table 2). While we cannot conclusively rule out other factors, the available evidence suggests that the decline in the Antipodes Islands population is most likely driven by ongoing environmental stressors rather than significant emigration.

We posit that the apparent population structure is likely at least partially explained by allochrony in their breeding cycles that results in each not significantly overlapping while at sea, at least during the pre-molt period [42]. In short, the STF sits just to the north of the Bounty Islands bringing high primary productivity to the region, likely negating the need

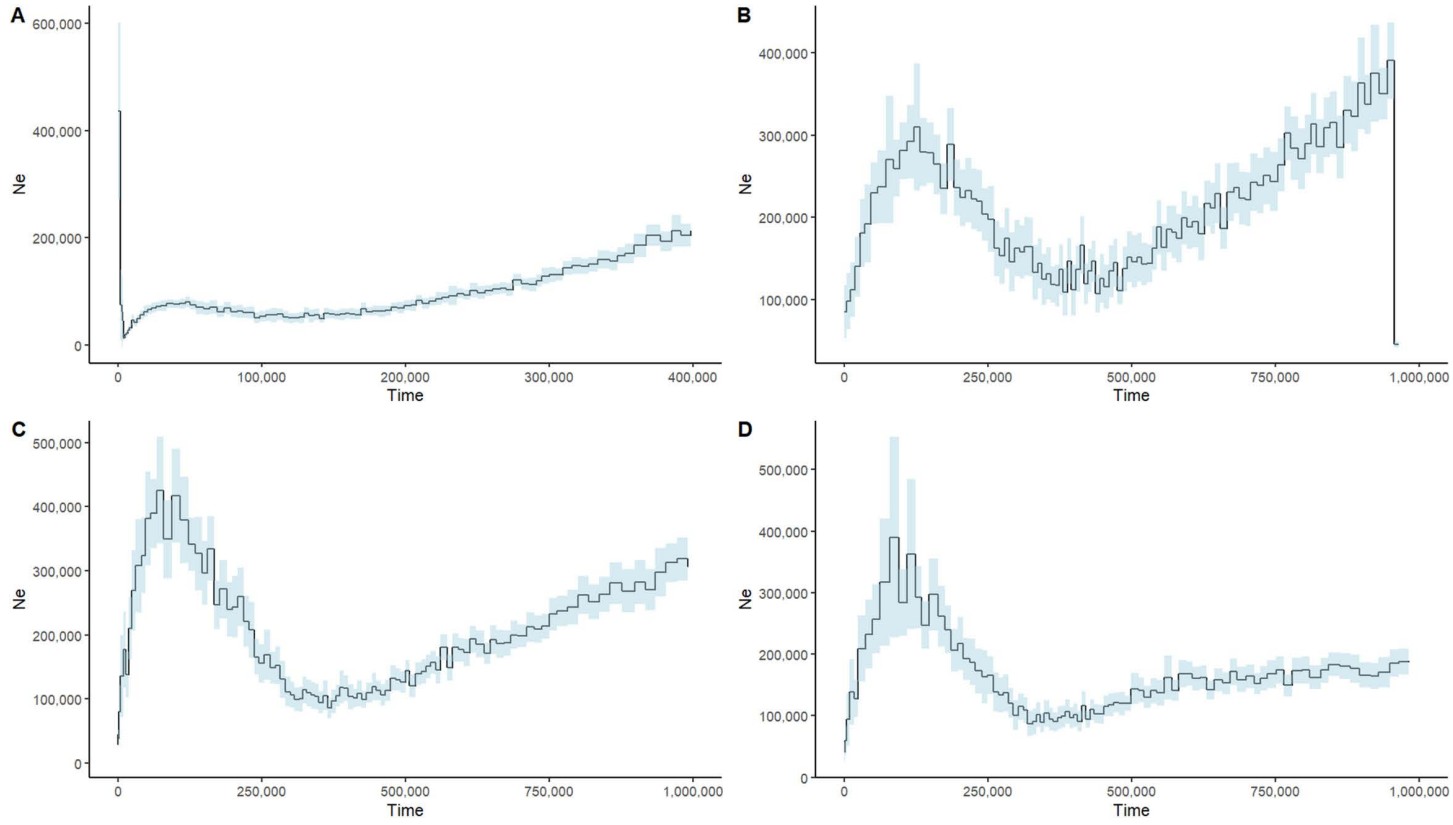

**Fig 6. Estimates of changes in historical effective populations size (*Ne*).** Historical changes to the effective population size (*Ne*) of A) tawaki, B) eastern rockhopper penguin, C) Antipodes Islands erect-crested penguins, and D) Bounty Islands erect-crested penguins. Shading represents the 95% confidence interval for the *Ne* estimates.

for foraging penguins to travel long distances in search of sufficient prey. Effectively utilizing different marine habitats (Bounty Islands = subtropical, Antipodes Islands = subantarctic) appears to contribute to a delayed breeding cycle on the Bounty versus Antipodes Islands. The Snares penguin (*Eudyptes robustus;* 'tawaki nana ho') also exhibits an asynchronous annual cycle with a 2–5 week delay in breeding onset between colonies on the main Snares group and the Western Chain [73–75]. Yet there are no differences in proximity to major oceanic fronts that could help explain this difference as these two island groups are only about 5 km apart. Other underlying factors must be influencing these behavioral differences in Snares penguins and therefore should not be ruled out in understanding similar patterns in erect-crested penguins.

The presence of frontal zones has been identified as a barrier to gene flow in other penguin species [17,19]. However, in the case of erect-crested penguins, the STF does not lie between the two island groups. Yet the differences in primary productivity fueled by the STF is likely an important factor resulting in behavioral differences which lead to isolation between these two populations. Most crested penguin species travel south during the pre-molt period [31,76,77], and the erect-crested penguins on the Antipodes Islands follow this pattern [42]. However, on the Bounty Islands, erect-crested penguins have shorter foraging ranges and travel generally north towards the STF, bucking this trend [42]. It could be that the presence of highly productive waters along the STF so close to their breeding islands prevents typical dispersal patterns in erect-crested penguins from the Bounty Islands, while those on the Antipodes Islands have retained the ancestral routines to travel south (Keys et al., in prep.).

## Conclusion

Our study expands the understanding of population structure and genetic connectivity for three species of crested penguins in New Zealand. Although closely related, each species exhibits unique ecology and population trends. Our results support previous findings that tawaki represent a single panmictic population across their range and that there is no significant structure among colonies or regions. Therefore, conservation and management efforts prioritizing the protection of marine habitats (both during and outside of the breeding season) and the mitigation of interactions with fisheries should continue.

In contrast, our results show that erect-crested penguins should be treated as two distinct populations corresponding to the Antipodes Islands and the Bounty Islands. There is likely some limited gene flow from the Antipodes Islands to the Bounty Islands, but our data do not suggest that the reverse occurs. The differences in the timing of breeding, foraging behavior, population trends, and genetics noted between these two island groups suggests that a "one size fits all" approach to the conservation of erect-crested penguins is not sufficient. Instead, the two island populations should be considered independently, with the Antipodes Island population likely needing more targeted research to identify the causes for their declines.

Eastern rockhopper penguins on the Antipodes Islands have very limited genetic diversity due to the continued decline in their population and little gene flow from other island groups. However, whether this low genetic diversity and lack of gene flow is true on other island groups (i.e., Campbell Island, Auckland Islands) is not well known. While the declines of eastern rockhoppers on the Antipodes Islands warrants continued research into the root causes of their decline, it may not present the full picture for this species. Therefore, we encourage further research into the genetic diversity, population structure, and ecology of eastern rockhoppers from other known breeding islands in the region (i.e., Campbell Island, Auckland Islands, Macquarie Island) to evaluate the status of this species in the Pacific Ocean.

## Supporting information

**S1 Table. List of all samples and sampling locations included.** Full list of tawaki (*Eudyptes pachyrhynchus*), erect-crested penguin (*Eudyptes sclateri*), and eastern rockhopper penguin (*Eudyptes filholi*) samples included in the analysis. (DOCX)

## Acknowledgments

We would like to thank Andrea Faris, Jacob Barrett, Daniel Crook, Holly Langley (Southern Discoveries), and Rosco Gaudin (Rosco's Kayaks) for providing logistic support to access colonies in Milford Sound. Access to Doubtful Sound colonies was made possible by Richard 'Abbo' and Mandy Abernathy's invaluable support (Fiordland Expeditions). We would also like to say a huge thank you to skipper Steve Kafka and the entire crew of the research vessel *Evohe* for providing transport and support into the Subantarctic islands. A massive thanks to the many field and lab assistants that have helped on this project including Blake Hornblow, Myrene Otis, Lindsey Chan, Briana Gibbs, Hannah Mattern, and Bianca Keys. Special thanks to the Department of Conservation. Particularly Sharon Trainor, Janice Kevern, and Rhuaridh Hannan for quarantine help, Ros Cole and Joseph Roberts for help with entry permits, and Graeme Taylor, Igor Debski, Johannes Fischer, Kris Ramm, Hendrik Schultz, and Sanjay Thakur for their help obtaining the necessary Wildlife Authority.

## Author contributions

**Conceptualization:** Jeff White, Thomas Mattern, Kevin G. McCracken.

**Data curation:** Jeff White, Philip Lavretsky, Alexis Diaz, Kevin G. McCracken.

**Formal analysis:** Jeff White, Philip Lavretsky, Alexis Diaz, Kevin G. McCracken.

**Funding acquisition:** Jeff White, Pablo Garcia Borboroglu, Thomas Mattern, Herman L. Mays Jr, Klemens Putz, Kevin G. McCracken.

**Investigation:** Jeff White, Ursula Ellenberg, David Houston, Robin Long, Thomas Mattern, Klemens Putz.

**Methodology:** Jeff White, Philip Lavretsky, Thomas Mattern, Herman L. Mays Jr, Kevin G. McCracken.

**Validation:** Philip Lavretsky, Kevin G. McCracken.

**Writing – original draft:** Jeff White.

**Writing – review & editing:** Jeff White, Philip Lavretsky, Pablo Garcia Borboroglu, Alexis Diaz, Ursula Ellenberg, David Houston, Robin Long, Thomas Mattern, Herman L. Mays Jr, Klemens Putz, Philip J. Seddon, Kevin G. McCracken.

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
