## [Decision Letter · Decision Letter 0]

30 Oct 2024

PONE-D-24-41700Population structure of three New Zealand crested penguins identifies current conservation challenges: Fiordland penguin/tawaki, erect-crested penguin, and Eastern rockhopper penguinPLOS ONE

Dear Dr. White,

Thank you for submitting your manuscript to PLOS ONE. After careful consideration, we feel that it has merit but does not fully meet PLOS ONE’s publication criteria as it currently stands. Therefore, we invite you to submit a revised version of the manuscript that addresses the points raised during the review process.

We look forward to receiving your revised manuscript.

Kind regards,

Vitor Hugo Rodrigues Paiva, Ph.D.

Academic Editor

PLOS ONE

Journal Requirements: When submitting your revision, we need you to address these additional requirements. 1. Please ensure that your manuscript meets PLOS ONE's style requirements, including those for file naming. The PLOS ONE style templates can be found at https://journals.plos.org/plosone/s/file?id=wjVg/PLOSOne_formatting_sample_main_body.pdf and https://journals.plos.org/plosone/s/file?id=ba62/PLOSOne_formatting_sample_title_authors_affiliations.pdf 2. To comply with PLOS ONE submissions requirements, in your Methods section, please provide additional information regarding the experiments involving animals and ensure you have included details on methods of anesthesia and/or analgesia. 3. Please include a complete copy of PLOS’ questionnaire on inclusivity in global research in your revised manuscript. Our policy for research in this area aims to improve transparency in the reporting of research performed outside of researchers’ own country or community. The policy applies to researchers who have travelled to a different country to conduct research, research with Indigenous populations or their lands, and research on cultural artefacts. The questionnaire can also be requested at the journal’s discretion for any other submissions, even if these conditions are not met.  Please find more information on the policy and a link to download a blank copy of the questionnaire here: https://journals.plos.org/plosone/s/best-practices-in-research-reporting. Please upload a completed version of your questionnaire as Supporting Information when you resubmit your manuscript 4. When completing the data availability statement of the submission form, you indicated that you will make your data available on acceptance. We strongly recommend all authors decide on a data sharing plan before acceptance, as the process can be lengthy and hold up publication timelines. Please note that, though access restrictions are acceptable now, your entire data will need to be made freely accessible if your manuscript is accepted for publication. This policy applies to all data except where public deposition would breach compliance with the protocol approved by your research ethics board. If you are unable to adhere to our open data policy, please kindly revise your statement to explain your reasoning and we will seek the editor's input on an exemption. Please be assured that, once you have provided your new statement, the assessment of your exemption will not hold up the peer review process. 5. Your ethics statement should only appear in the Methods section of your manuscript. If your ethics statement is written in any section besides the Methods, please move it to the Methods section and delete it from any other section. Please ensure that your ethics statement is included in your manuscript, as the ethics statement entered into the online submission form will not be published alongside your manuscript.

Reviewers' comments:

Reviewer's Responses to Questions

**Comments to the Author**

1. Is the manuscript technically sound, and do the data support the conclusions?

Reviewer #1: Yes

Reviewer #2: Yes

Reviewer #3: Yes

Reviewer #4: Partly

2. Has the statistical analysis been performed appropriately and rigorously? 

Reviewer #1: Yes

Reviewer #2: Yes

Reviewer #3: Yes

Reviewer #4: Yes

3. Have the authors made all data underlying the findings in their manuscript fully available?

Reviewer #1: Yes

Reviewer #2: No

Reviewer #3: No

Reviewer #4: Yes

4. Is the manuscript presented in an intelligible fashion and written in standard English?

Reviewer #1: Yes

Reviewer #2: Yes

Reviewer #3: Yes

Reviewer #4: Yes

5. Review Comments to the Author

Reviewer #1: Overall the presented study adds to our current understanding of the genetic diversity and gene flow of the three species studied. The paper is nicely written, well laid out and analysed. I have some detailed comments on specific areas below:

Abstract – line 40 and 41 you introduce the species you are working on in this paper, but the next lines use tawaki to describe what was introduced as Fiordland, and adds sub-Antarctic to erect crested. I found it a little confusing and had to read the two again, so perhaps alter to improve consistency.

Introduction

Under study system lines 81-82, you detail what the study is about, however, be clear that yours is looking at genetic population structure, and I do not think your study is looking at population trends in population size – please be clear that you are discussing genetic population structure and gene flow – as this paper cannot possible look at all these things together. You use historical published accounts to make sense of your genetic findings, but you do not present population trend data for example.

Figure 1 map – why are the two species on the Antipodes islands in the same colour? I find the key confusing – the colours are for locations, but then you have listed species.

Fig 2. I dislike the colours and the shapes here – they are not matching the key and are confusing. The red and blue are much darker on the plot than the key and the Eastern Rockhopper star shape is also a different blue to the island colour in the key. Spelling error of Eastern Rockhopper in the key (Roockhopper).

Conclusion: Something is a little lacking in the conclusion – perhaps further details on how or what researchers could investigate to understand why the Antipodes population is in such decline, or what additional genetic research could add to our understanding or aiding species recovery. Perhaps relate back to the introduction and initial aims and reasoning as to why you carried out the study in the first instance. I just feel it could be stronger to emphasise the important results you have found.

Reviewer #2: This is a very important research regading genetic diversity of three penguin species using a robust genetic technique and a well-performed bionformatics pipeline.

There are some important missing details in the introduction and methodology, and some of the paragraphs of the discussion must be improved in order to clarify the authors' ideas. All my considerations were inserted as notes in the marked pdf.

Reviewer #3: The manuscript entitled ‘Population structure of three New Zealand crested penguins identifies current conservation challenges: Fiordland penguin/tawaki, erect-crested penguin, and Eastern rockhopper penguin’ investigated the population structure of three related crested penguin species in New Zealand using genome wide SNP data. Results showed panmixia among tawaki colonies, limited genetic diversity in eastern rockhopper penguins and significant genetic structure between two populations of erect-crested penguin exhibiting different patterns of foraging behaviour.

This study has a well-conceived biological question, especially regarding the long-distance movable seabirds. The data produced is astutely analyzed and the figures are well designed even though some of them could be improved (see further comments). Yet I have some concerns about the interpretation of the results. I have commented at greater length on this matter below and I think those ambiguities should be fixed and further clarified before the final acceptance of the paper.

Introduction

L62 – Please reword for clarity: (…) across all groups of seabirds and penguins are particularly vulnerable (…).

L64-L66 – The author introduced penguins as ‘particularly vulnerable species’ in L60-64 then started a new sentence with the importance of understanding the ecology and evolutionary of species. I would recommend replacing ‘species’ with ‘penguins’.

L64-L66 – Replace ‘vital’ with necessary/required or important.

L77-L79 – Please develop the case of king penguins. It is a wide-ranging seabird species, so the example may relate to IBD, but is there an overlap in foraging areas among populations? It is not clear how this example is related to the above descriptions of putative mechanisms of genetic divergence among seabird populations.

Study system

L81-L113 – The problematic is well explained and clear!

L119 – Please define ‘genetic histories’. Do you mean evolutionary histories?

L122 – Please replace ‘unstructured’ with ‘panmictic’.

L123 – Please replace ‘decreased’ with ‘low genetic diversity’.

L121-122 – Add at the end of the sentence that the genetic markers used in the present study are different from the mitochondrial DNA locus used in a previous study on the same species otherwise the reader does not understand how this study is different from the analyzes that have already been conducted (same in the section Conclusion).

Material and methods

L136 – For more clarity, : (…) of 2017, 2018, and 2022 divided into four regions: (…), and Foveaux Strait (Whenua Hou) (Fig1; S1 Table).’

DNA extraction and ddRAD-Seq library preparation

L155-L157 – It would be of interest to indicate which DNA samples were extracted with the DNeasy Blood and Tissue kit or with phenol-chloroform-isoamyl alcohol extractions.

Results

Population structure/Summary statistics

Admixture plots, PCA, FineRADstructure heatmaps and summary statistics ɸST allowed coancestry assignments and detection of potential genetic clustering among populations and species. However, they do not take into consideration demographic histories of species – i.e. population trajectories or population trends. However, the author mentions several times those aspects throughout the manuscript (e.g. in the section Conclusion L366). I would find it interesting to build models that would give an idea of the evolution of the effective population size of populations (Ne) through time as it may greatly affect the results – especially for erect-crested penguins. Indeed, while gene flow can be inferred from the amount of neutral allele frequency divergence among populations, deviation from the ‘island model’ of population structure following bottlenecks or population expansion may result in erroneous representation of contemporary gene flow based on population genetic clustering analyses.

In Table 2 & 3, it is not clear whether ɸST and Tajima’D are significant or not.

L269 - L270 – The author argues that the high degree of coancestry might indicate a limited gene pool likely due to historical population declines. This implies that the populations might not be at mutation-drift equilibrium and hence that the results might be biased. This must be clarified.

In Fig 2 & 4 Symbols in plots could be bigger to help the reader.

Discussion

L289-L290 – The author concludes that population structure among tawaki populations is unlikely masked by recent population expansions. According to the previous considerations, the author should also conclude or at least mention that the tawaki case study does not support monogamy and philopatry as factors of genetic differentiation among seabird populations.

L318-L320 – I would discard this sentence as the author addresses it in L323-L326.

L331-L340 – I would be careful with the concluding remarks. The observation of highly structured erect-crested penguin populations might also reflect the size reduction of the antipode population. Current gene flow between colonies might occur but not be sufficient to overridden dramatic demographic history. The sample sizes for this species for Antipodes Islands are: Anchorage Bay N=14; South Coast N=14, which is quite low. I would recommend developing the demographic history aspect of erect-crested penguins in parallel to the allochrony explanation in the section discussion.

Reviewer #4: This manuscript investigates genetic connectivity, structure, and diversity in three penguin species in New Zealand: Fiordland penguin (tawaki), erect-crested penguin, and Eastern rockhopper penguin. Using ddRADseq SNP data, the authors examined population structure across different colonies, focusing on conservation implications given observed population declines.

The article is very well-written, with clear background on each of the species and a clear set of testable hypotheses to be explored presented in the introduction and finishes with a nicely rounded discussion. The genetic results support known demographics of the species and populations. I have a few comments which I think will improve the statistical validity and interpretation of the results. I recommend additional analyses, for which the authors can decide how necessary they are. However, I think with such a nice dataset, pushing it just a little further in regards to such analyses, will help support the author’s key messages.

Main comments:

Line 202: fineRADstructure uses the full haplotype information (i.e. every SNP on the RAD locus). Please confirm whether all SNPs were used or if the LD pruned dataset (described previously) was used. This can quite dramatically change the results of fineRADstructure so please be sure of this.

Line 268-269 (also lines 308-309). I would recommend checking via other methods (e.g. vcftools –relatedness2 option) to see if these two samples really do represent siblings. Looking at your fineRADstructure plot, it looks to me as though there may be other related individuals in this sample too. Related to this, you should remove one member of the any related pair before calculating the pop. gen summary statistics.

Line 232: I can’t see obvious “hybrids” in your PCA. Please elaborate.

In Fig. 4 (PCA of erect-crested penguins), it’s curious to me that PC1 (14%) separates individuals sampled from both colonies, whereas it is PC2 (6.8 %) that differentiates the colonies. What’s going on along PC1? Are the samples lower-coverage? Lower genetic diversity? Please investigate these outlier samples. If there is not data quality reason to remove these individuals, then please discuss this result.

Line 270: yes to population declines, but also isolation and genetic drift (see my comments below regarding gene flow and effective population size analyses).

In the fineRADstructure plot, is there a reason for the Eastern rockhopper sharing higher ancestry with the Erect-crested penguin? This would be worth discussing

Given the large weight placed on gene flow between colonies placed in the discussion, a formal analysis of gene flow would be a nice addition. i.e. using BayesAss (https://github.com/brannala/BA3). See https://onlinelibrary.wiley.com/doi/full/10.1111/ddi.13399 for a nice use of this in rockhopper penguins. Such an analysis is relatively straightforward and would add an analytical framework to much of what is currently discussed.

I would also love to see estimates of the effective population size, especially for tawaki and erect-crested penguins as the sample sizes are more than high enough for performing these estimates. Providing these estimates would substantially help the author’s claims about the different changes in demography and in interpreting the population structure and genetic diversity estimates. Using for example: https://onlinelibrary.wiley.com/doi/10.1111/1755-0998.13890

Minor comments:

In the keywords, I would suggest using ddRADseq (capitalised RAD) as in the rest of the manuscript.

Line 45: instead of “thousands of autosomal loci” can you be more quantitative?

Line 171: cite the genome paper? https://doi.org/10.1093/gigascience/giz117

Line 171-172. You probably used the mem algorithm of bwa. Specify this

Line 200: hyphenate “species-specific”

Line 263: “lower differentiation”

Line 270: remove the word “Calculating”

I would recommend using a fewer number of decimal places for some of the statistics reported in Table 2 to increase readability.

The Figures are very low resolution, meaning I couldn’t interpret, for example, the PCA presented in Figure 2. Please replace with a higher-resolution figure.

6. PLOS authors have the option to publish the peer review history of their article (what does this mean? ). If published, this will include your full peer review and any attached files.

**Do you want your identity to be public for this peer review?** For information about this choice, including consent withdrawal, please see our Privacy Policy .

Reviewer #1: No

Reviewer #2: No

Reviewer #3: No

Reviewer #4: No

---

## [Author Response · Author response to Decision Letter 1]

6 May 2025

We thank the reviewers for their helpful comments. We believe that the revised manuscript is stronger after incorporating their comments. We have provided a detailed response to each comment in the supplied Response to Reviewers document.

---

## [Decision Letter · Decision Letter 1]

29 May 2025

PONE-D-24-41700R1Population structure of three New Zealand crested penguins identifies current conservation challenges for the Fiordland penguin/tawaki, erect-crested penguin, and Eastern rockhopper penguinPLOS ONE

Dear Dr. White,

Thank you for submitting your manuscript to PLOS ONE. After careful consideration, we feel that it has merit but does not fully meet PLOS ONE’s publication criteria as it currently stands. Therefore, we invite you to submit a revised version of the manuscript that addresses the points raised during the review process.

We look forward to receiving your revised manuscript.

Kind regards,

Vitor Hugo Rodrigues Paiva, Ph.D.

Academic Editor

PLOS ONE

Journal Requirements:

Reviewers' comments:

Reviewer's Responses to Questions

**Comments to the Author**

1. If the authors have adequately addressed your comments raised in a previous round of review and you feel that this manuscript is now acceptable for publication, you may indicate that here to bypass the “Comments to the Author” section, enter your conflict of interest statement in the “Confidential to Editor” section, and submit your "Accept" recommendation.

Reviewer #2: (No Response)

Reviewer #3: All comments have been addressed

2. Is the manuscript technically sound, and do the data support the conclusions?

Reviewer #2: Yes

Reviewer #3: Yes

3. Has the statistical analysis been performed appropriately and rigorously? 

Reviewer #2: Yes

Reviewer #3: Yes

4. Have the authors made all data underlying the findings in their manuscript fully available?

Reviewer #2: No

Reviewer #3: Yes

5. Is the manuscript presented in an intelligible fashion and written in standard English?

Reviewer #2: Yes

Reviewer #3: Yes

6. Review Comments to the Author

Reviewer #2: The authors mostly addressed the suggestions, corrections and questions of the reviewers. I believe this is a well-conducted study on genetic diversity and connectivity that highlights the importance of conservation measures directed toward evolutionarily significant units; for these reasons, it is worthy of publication in PLOS One.

Some final considerations:

In L49-50, I think that Eastern rockhopper penguin should be mentioned beside the erected-crested penguins. In the next sentence, when you speak directly of the Eastern rockhopper population of Antipodes Islands, it seems disconnected because you haven’t mentioned the other species previously.

L81-82: I agree with Reviewer #3 when he stated that “it is is not clear how this example is related to the above descriptions of putative mechanisms of genetic divergence among seabird populations.” Despite the rewording of this sentence, I still feel like it is disconnected to the aforementioned possibilities. The authors stated that “We suggest in the results that dispersal ability in tawaki is an important factor in the lack of structure across their range (similar to the king penguins) and a lack of dispersal is likely a contributing factor in the decline of Eastern rockhopper populations and low genetic diversity on Antipodes Island.” I understand the connection of the example of the King Penguins to the authors’ results, but it doesn’t aggregate much to mention a different species in which this happens before the reader even knows what your results are going to be (in the introduction). I think that maybe the authors should just mention this possibility as just another one, without giving it a highlight with the King Penguins example.

In L132-134, the authors state that “Finally, we investigated whether the presence of the STF, and associated higher primary productivity, influences gene flow between the Antipodes Islands and Bounty Islands populations of erect-crested penguins.” I am concerned that the authors did not, in fact, investigate the influence of different levels of primary productivity on the genetic structuring between the two populations. Rather, they discuss the possibility of this phenomenon, providing examples of other species in which this has been observed. In order to investigate the influence of primary productivity as a selective pressure, authors should have obtained chlorophyll-α levels in the foraging areas of each population, and then tested this association (or influence) through Mantel tests or Latent Factor Mixed Models (LFMMs), for example. Some examples of works that did, in fact, test this association are Nunes & Bugoni (2017) - https://onlinelibrary.wiley.com/doi/abs/10.1111/jbi.13142 ; Mazzochi et al. (2024) – https://link.springer.com/article/10.1007/s10592-024-01613-x .

If the authors are willing to do this, I believe it would be an interest analysis that could provide a more robust answer to their research question. However, if they are not, the referenced sentence should be reworded – avoiding the phrasing that frames this as an analysis, and instead using terms such as “discussed”, “considered the possibility”, or similar.

L352-353: I still don’t understand why the authors mention that population size and genetic diversity would rebound if there were modest levels of immigration. Are they arguing that there is, in fact, no immigration? But what evidences show that there is/there is not immigration? As they give the example of the king penguins that recovered pre-bottleneck levels and then don’t make a link with the study case, I don’t understand why they mention the importance of immigration for this situation.

In L362-364, the authors also say that a little gene flow could be existing from other colonies to the Antipodes Islands, which reduces chances of genetic recovery. But shouldn’t it increase the chances, as the authors previously stated? Or have they made an attempt to argue that this “little gene flow” wouldn’t be enough to rebound pre-bottleneck levels of genetic diversity?

Finally, it appears that the authors have not provided a link to the available genomic data in the manuscript (data availability statement?). I believe this is an important detail that should be included.

Reviewer #3: The authors have adequately addressed my comments raised in a previous round of review and I feel that this manuscript is now acceptable for publication

7. PLOS authors have the option to publish the peer review history of their article (what does this mean? ). If published, this will include your full peer review and any attached files.

**Do you want your identity to be public for this peer review?** For information about this choice, including consent withdrawal, please see our Privacy Policy .

Reviewer #2: No

Reviewer #3: **Yes: ** Anicée Lombal

---

## [Author Response · Author response to Decision Letter 2]

8 Jul 2025

Response to Reviewer Comments

PONE-D-24-41700R1

Population structure of three New Zealand crested penguins identifies current conservation challenges for the Fiordland penguin/tawaki, erect-crested penguin, and Eastern rockhopper penguin

We would like to thank the reviewers for their time and many helpful comments throughout this review process. Please find our comments to the suggested revisions below.

Review Comments to the Author

Reviewer #2: The authors mostly addressed the suggestions, corrections and questions of the reviewers. I believe this is a well-conducted study on genetic diversity and connectivity that highlights the importance of conservation measures directed toward evolutionarily significant units; for these reasons, it is worthy of publication in PLOS One.

Some final considerations:

In L49-50, I think that Eastern rockhopper penguin should be mentioned beside the erected-crested penguins. In the next sentence, when you speak directly of the Eastern rockhopper population of Antipodes Islands, it seems disconnected because you haven’t mentioned the other species previously.

We thank the reviewer for this comment, however we do not agree that the eastern rockhopper should also be mentioned in this sentence. This statement is referencing our finding of population structure in erect-crested penguins (only) based on breeding island group. We did not include eastern rockhoppers in this same statement as they do not occur on both island groups and therefore mentioning them in this sentence could be misleading. This structure is the primary finding we wish to convey about erect-crested penguins while low genetic diversity is the primary finding for eastern rockhoppers. We also do mention eastern rockhoppers earlier in the abstract in the same (and only) sentence that introduced the erect-crested penguins.

L81-82: I agree with Reviewer #3 when he stated that “it is is not clear how this example is related to the above descriptions of putative mechanisms of genetic divergence among seabird populations.” Despite the rewording of this sentence, I still feel like it is disconnected to the aforementioned possibilities. The authors stated that “We suggest in the results that dispersal ability in tawaki is an important factor in the lack of structure across their range (similar to the king penguins) and a lack of dispersal is likely a contributing factor in the decline of Eastern rockhopper populations and low genetic diversity on Antipodes Island.” I understand the connection of the example of the King Penguins to the authors’ results, but it doesn’t aggregate much to mention a different species in which this happens before the reader even knows what your results are going to be (in the introduction). I think that maybe the authors should just mention this possibility as just another one, without giving it a highlight with the King Penguins example.

We have removed the mention of king penguins here and replaced the citation with a more general metanalysis instead to provide a broader example of the mechanism at this stage of the paper.

In L132-134, the authors state that “Finally, we investigated whether the presence of the STF, and associated higher primary productivity, influences gene flow between the Antipodes Islands and Bounty Islands populations of erect-crested penguins.” I am concerned that the authors did not, in fact, investigate the influence of different levels of primary productivity on the genetic structuring between the two populations. Rather, they discuss the possibility of this phenomenon, providing examples of other species in which this has been observed. In order to investigate the influence of primary productivity as a selective pressure, authors should have obtained chlorophyll-α levels in the foraging areas of each population, and then tested this association (or influence) through Mantel tests or Latent Factor Mixed Models (LFMMs), for example. Some examples of works that did, in fact, test this association are Nunes & Bugoni (2017) - https://onlinelibrary.wiley.com/doi/abs/10.1111/jbi.13142 ; Mazzochi et al. (2024) – https://link.springer.com/article/10.1007/s10592-024-01613-x .

If the authors are willing to do this, I believe it would be an interest analysis that could provide a more robust answer to their research question. However, if they are not, the referenced sentence should be reworded – avoiding the phrasing that frames this as an analysis, and instead using terms such as “discussed”, “considered the possibility”, or similar.

We thank the reviewer for this suggestion. While we do agree that adding environmental data (chlorophyll-a and SST especially) would be very interesting, we believe it is out of the scope and desired purpose for this paper. However, we are working on another paper that does look more in depth into the divergent ecology of the erect-crested penguin populations. Therefore, we have updated the wording of this sentence to avoid unnecessary confusion.

L352-353: I still don’t understand why the authors mention that population size and genetic diversity would rebound if there were modest levels of immigration. Are they arguing that there is, in fact, no immigration? But what evidences show that there is/there is not immigration? As they give the example of the king penguins that recovered pre-bottleneck levels and then don’t make a link with the study case, I don’t understand why they mention the importance of immigration for this situation.

In this case we are making the point that the eastern rockhopper population on Antipodes is likely very isolated. If there was immigration from other populations of eastern rockhoppers (i.e., from Campbell, Auckland, or Macquarie) then we would expect genetic diversity to be higher. But what we do need is sampling from other populations to quantify gene flow among them and assess the status of this species at other sites. This is exactly the focus of our expanded work onto these islands starting in the 2025 field season. We have reworded the line to clarify.

In L362-364, the authors also say that a little gene flow could be existing from other colonies to the Antipodes Islands, which reduces chances of genetic recovery. But shouldn’t it increase the chances, as the authors previously stated? Or have they made an attempt to argue that this “little gene flow” wouldn’t be enough to rebound pre-bottleneck levels of genetic diversity?

We appreciate the reviewer pointing out this confusing wording. By “little gene flow” we meant that there is potentially a lack of gene flow, not that there is a small amount of gene flow. We have updated for clarity.

Finally, it appears that the authors have not provided a link to the available genomic data in the manuscript (data availability statement?). I believe this is an important detail that should be included.

We have uploaded the data to the NCBI SRA repository to be available upon publication at BioProject ID PRJNA1277195.

Reviewer #3: The authors have adequately addressed my comments raised in a previous round of review and I feel that this manuscript is now acceptable for publication

We thank the reviewer for their comments and efforts to help us improve the quality of this paper.

---

## [Editor Report · Decision Letter 2]

18 Jul 2025

Population structure of three New Zealand crested penguins identifies current conservation challenges for the Fiordland penguin/tawaki, erect-crested penguin, and eastern rockhopper penguin

PONE-D-24-41700R2

Dear Dr. White,

We’re pleased to inform you that your manuscript has been judged scientifically suitable for publication and will be formally accepted for publication once it meets all outstanding technical requirements.

Kind regards,

Vitor Hugo Rodrigues Paiva, Ph.D.

Academic Editor

PLOS ONE
---

## [Editor Report · Acceptance letter]

PONE-D-24-41700R2

PLOS ONE

Dear Dr. White,

I'm pleased to inform you that your manuscript has been deemed suitable for publication in PLOS ONE. Congratulations! Your manuscript is now being handed over to our production team.

Kind regards,

on behalf of

Dr. Vitor Hugo Rodrigues Paiva

Academic Editor

PLOS ONE